# PromptBoosting: Black-Box Text Classification with Ten Forward Passes

## Abstract

We describe PromptBoosting, a query-efficient procedure for building a text classifier from a neural language model (LM) without access to the LM's parameters, gradients, or hidden representations. This form of "black-box" classifier training has become increasingly important as the cost of training and inference in large-scale LMs has grown. But existing black-box LM classifier learning approaches are themselves computationally inefficient, typically specializing LMs to the target task by searching in a large space of (discrete or continuous) prompts using zeroth-order optimization methods. Instead of directly optimizing in prompt space, PromptBoosting obtains a small pool of prompts via a gradient-free approach, and then constructs a large pool of *weak learners* by pairing these prompts with different elements of the LM's output distribution. These weak learners are then ensembled using the AdaBoost algorithm. The entire learning process requires only a small number of forward passes per batch and no backward pass. Experiments show that PromptBoosting achieves state-of-the-art performance in multiple black-box few-shot classification tasks, and matches or outperforms full fine-tuning in both few-shot and standard learning paradigms, while training 10x faster than existing black-box methods.

## 1 Introduction

Prompt-based learning has emerged as an effective method to adapt pretrained language models (LMs) for downstream natural language processing (NLP) tasks. A typical prompt-learning paradigm involves appending a specially-designed sequence, called a prompt, to the input to a pretrained LM, which will thereby be repurposed for a given downstream task. Compared to the standard fine-tuning, prompt-based learning is much more parameter-efficient.

Most prompt-based learning methods require searching for the optimal prompt for the downstream task. When gradient information of the pre-trained LM is available, such optimization can easily be performed by standard gradient-based methods (Liu et al., 2021; Li & Liang, 2021; Lester et al., 2021; Zhang et al., 2021; Liu et al., 2022). However, in many real-world scenarios, the parameters, gradient or hidden representations of the LMs are not accessible, *a.k.a.* the black-box tuning setting, which makes gradient-based prompt learning very challenging (Sun et al., 2022).

To tackle the challenges, the most common existing black-box solution is to resort to gradient-free optimization techniques to search for the optimal prompt, such as the zeroth-order gradient approximation (Sun et al., 2022; Diao et al., 2022) and reinforcement learning-guided optimization (Deng et al., 2022). However, these methods would require a large number of queries of the LMs, which, considering the ever-growing size and computation cost of the pre-trained LMs, is highly inefficient and could lead to large approximation errors.

In this paper, we propose PromptBoosting, a novel black-box prompt learning approach which does not rely on searching an optimal prompt, and which can thus drastically improve the computational efficiency over the existing methods. Figure 1 illustrates the pipeline of PromptBoosting. Specifically, rather than optimizing over the prompts, PromptBoosting constructs a small pool of prompts via a gradient-free approach. These prompts are sub-optimal because they are not optimized for any downstream tasks. Then, PromptBoosting creates a large pool of *weak learners* by pairing each prompt with different elements of the LM's output distribution, which is commonly

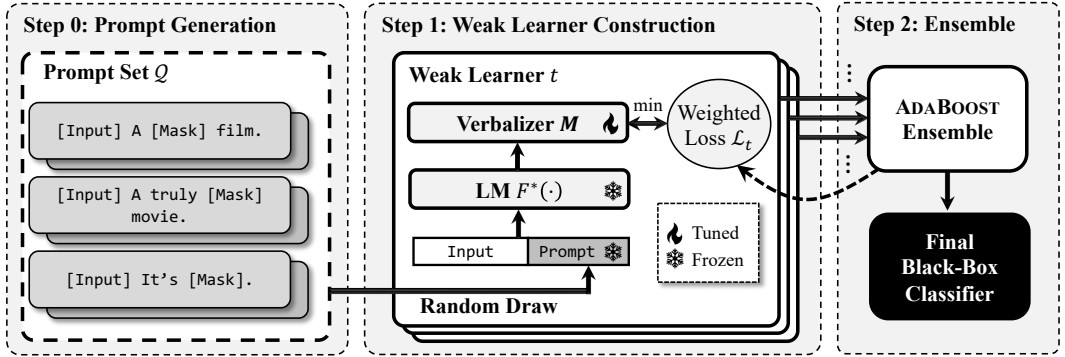

Figure 1: Overview of PROMPTBOOSTING.

known as the *verbalizer*. Finally, these weak learners are ensembled using the ADABOOST algorithm, where the optimization in each iteration is performed only over the verbalizer, not the prompt. The entire process only needs to evaluate the LM's output with each of weak prompts, so it only involves a small number of forward passes per batch, and no backward pass.

We evaluated our method on a number of downstream tasks and LM's. The results show that PROMPTBOOSTING achieves state-of-the-art performance and matches or even outperforms full fine-tuning in both few-shot and standard learning paradigms. Furthermore, PROMPTBOOSTING can run 10x faster than existing black-box prompt-learning approaches, with only ten forward passes per batch. PROMPTBOOSTING can inspire a new black-box prompt learning paradigm.

## 2    RELATED WORK

**Prompt-based learning**    Prompt-based learning has emerged as a new approach for adapting pre-trained LMs for downstream tasks fueled by the success of GPT-3 (Brown et al., 2020). Since the prompts directly influence the performance of prompt-based learning, recent studies have focused on how to find the best prompts given a specific task. AutoPrompt (Shin et al., 2020) designs a gradient-based discrete optimization method to search for the optimal prompt. LM-BFF (Gao et al., 2021) leverages the pre-trained T5 model (Raffel et al., 2020) to automatically generate prompts and select the best one based on the performance on the validation set. Since verifying the automatically-generated prompts is still time-consuming, the PTR (Han et al., 2021) method incorporates logic rules to construct prompts and to encode prior knowledge into prompt-based learning.

Another line of work replaces the discrete prompt tokens with continuous embeddings that have their own parameters. P-tuning (Liu et al., 2021) trains a BiLSTM network to output continuous prompt embeddings. Prefix-tuning (Li & Liang, 2021) inserts prompt embeddings to each transformer layer in LMs and optimizes only the prompt embeddings during training. Prompt Tuning (Lester et al., 2021) also keeps the LMs frozen but adds prompt embeddings only in the input. In addition to adding prompt embeddings to each layer, P-tuning V2 (Liu et al., 2022) replaces the language model head in LMs with a linear layer for classification and shows that soft prompt tuning scales to medium-sized LMs and hard sequence tagging tasks. Our work adopts the automatically generated discrete prompts for prompt-based learning.

**Black-box Tuning**    Extremely large LMs such as GPT-3 are only provided as a service in the cloud, resulting inaccessible parameters and gradients of LMs. Furthermore, from the model provider's perspective, sharing hidden representations or gradients of LMs may reveal the vulnerability of the model and lead to security problems (Tramèr et al., 2016). How to find the optimal prompts in such black-box tuning setting has attracted various explorations. BBT (Sun et al., 2022) employs the CMA evolution strategy, a derivative-free optimization method, to optimize continuous prompt embeddings. However, BBT requires querying the LM tens of thousands of times even in few-shot settings. Furthermore, BBT's use of soft prompts violates the black-box tuning setting because in realistic use one can query the LM service with only textual input instead of prompt embeddings. Other concurrent work (Diao et al., 2022) also uses gradient-free techniques to optimize soft

prompts. RLPrompt (Deng et al., 2022) is a more realistic black-box tuning method where discrete prompt tokens are optimized through reinforcement learning and the performance on downstream tasks serves as the reward. GrIPS (Prasad et al., 2022) performs phrase-level editing to generate discrete prompts for downstream tasks. Existing black-box tuning methods suffer from poor efficiency and sub-optimal performance. In contrast, our method achieves high efficiency by first generating only a small set of prompts and achieves superior performance by then creating a set of weak learners from these prompts and ensembling them together via ADABOOST (Freund & Schapire, 1997).

**Model ensemble.** Model ensembling is a commonly used technique in machine learning. Prior to deep learning, Bagging (Breiman, 1996; 2001) and Boosting (Freund & Schapire, 1997; Friedman, 2001) showed the power of model ensembling. One of these methods, ADABOOST (Freund & Schapire, 1997), sequentially learns a series of weak learners and ensembles them for better generalization. During training, each weak learner is tweaked by leveraging examples that were misclassified by previous classifiers. Since the performance of each individual prompt can be weak, our method adopts ADABOOST as the framework for learning and ensembling multiple prompts.

**Prompt ensemble.** As has been pointed out by prior work (Lester et al., 2021), ensembling prompts is more efficient than ensembling entire fine-tuned models. Various ensemble strategies have been explored in past work. Uniformly averaging the predictions from different prompts has been used for factual probing (Jiang et al., 2020), text generation (Yuan et al., 2021; Schick & Schütze, 2020), and classification tasks (Schick & Schütze, 2021a; Lester et al., 2021). Furthermore, some methods adopt a weighted averaging strategy for better performance—the weight of each different prompt can be learned during training (Jiang et al., 2020; Qin & Eisner, 2021) or defined using some heuristics (Schick & Schütze, 2021a;b). Our method also falls into the prompt ensemble category. The main difference is each prompt-based model is sequentially learned conditioned on the classification errors of prior models.

## 3 METHODOLOGY

In this section, we will describe the PROMPTBOOSTING algorithm. For notation, we use $|\mathcal{A}|$ to denote the size of a finite set $\mathcal{A}$; $[A]$ to denote an index set $\{1, 2, \cdots, A\}$.

### 3.1 PROBLEM FORMULATION AND THE PROMPT-LEARNING FRAMEWORK

Consider a text classification downstream task. Denote $\mathcal{D}_{tr} = \bigcup_i \{(\boldsymbol{x}_i, y_i)\}$ as the training set, where $\boldsymbol{x}_i$ denotes the input text sequence and $y_i$ denotes the output label. We are given a pre-trained language model, denoted as $\boldsymbol{p}_i = F^*(\boldsymbol{x}_i)$, which, given the input $\boldsymbol{x}_i$, produces a probability distribution over the vocabulary set, $\mathcal{V}$, at a given location. In this paper, only the output distribution at where the input is [mask] is relevant, so $\boldsymbol{p}_i \in \mathbb{R}^{|\mathcal{V}| \times 1}$ is just a $|\mathcal{V}|$-dimensional vector specifying the output probability at the [mask] location, where $|\mathcal{V}|$ denotes the vocabulary size. Our goal is to adapt the LM $F^*(\cdot)$ to the downstream task using the downstream training set $\mathcal{D}_{tr}$.

We adopt the common prompt-learning framework, where the parameters of $F^*(\cdot)$ are frozen (hence we add a superscript $^*$ to emphasize this). Instead, the following two mechanisms are added to convert $F^*(\cdot)$ into a text classifier for the given downstream tasks.

1. **Prompt** A prompt is sequence of tokens that is concatenated to the input. Formally, denote the prompt sequence as $\boldsymbol{q}$ and the concatenated input sequence as $\boldsymbol{x}_i \| \boldsymbol{q}$. Then the LM is modified as $F^*(\boldsymbol{x}_i \| \boldsymbol{q})$.

2. **Verbalizer** To convert the output probability over the *vocabulary* into that over the *classes*, a verbalizer is introduced to assign each token into different classes. Formally, denote the number of classes of the downstream class as $|\mathcal{Y}|$, then the verbalizer is an $|\mathcal{Y}|$-by-$|\mathcal{V}|$ matrix, denoted as $\boldsymbol{M}$, where the element in row $c$, column $v$ represents the assignment weight of the $v$-token in the vocabulary into class $c$. Each row of $\boldsymbol{M}$ would sum up to one. The predicted probability of all the classes can then be expressed as $\boldsymbol{M}\boldsymbol{p}_i$.

To sum up, after the prompt and verbalizer are applied, the adapted LM becomes $MF^*(x_i\|q)$. Therefore, the prompt-tuning process boils down to learning an appropriate verbalizer $M$ and prompt $q$ for the downstream task.

## 3.2 ALGORITHM OVERVIEW

Conventional black-box prompt learning methods commonly use a pre-set $M$ while performing black-box optimization over $q$, which results in a large computation cost. On the contrary, PROMPT-BOOSTING randomly chooses from a small number of pre-generated prompts and performs optimization over $M$ instead. Due to the sub-optimality of pre-generated prompts and the limited representation power of $M$, the resulting classifiers are weak. However, this process is able to quickly generate a large pool of such weak learners, which can then be ensembled into a strong learner using the ADABOOST approach. Since the optimization over $M$ is computationally cheap, the ensembling process is still much more efficient than the conventional black-box methods.

More specifically, the ADABOOST algorithm iteratively generates $T$ weak learners, and weaker learner $t$ is optimized under its respective loss function, denoted as $\mathcal{L}_t(q, M)$, which is essentially a weighted loss over the training set with larger weights on those that are misclassified by the previous weak learners (More details of the ADABOOST algorithm will be provided in Section 3.5). Then, as shown in Figure 1, PROMPTBOOSTING consists of the following key steps.

**Step 0:** Generate a pool of prompts, $\mathcal{Q} = \bigcup_j \{q_j\}$, using a gradient-free method.

**Step 1:** Construct $T$ weak learners. For weak learner $t$, its prompt $q_t$ is uniformly randomly drawn from $\mathcal{Q}$; its verbalizer $M$ is determined by solving

$$\min_M \mathcal{L}_t(q_t, M), \quad \text{s.t.} \quad M_{cv} \geq 0, \forall c \in [[\mathcal{Y}]], v \in [[\mathcal{V}]], \quad \sum_{c \in [[\mathcal{Y}]]} M_{cv} = 1, \forall v \in [[\mathcal{V}]]. \tag{1}$$

**Step 2:** Ensemble the weak learners according to ADABOOST.

Section 3.3 will describe how to solve equation 1. Section 3.4 will describe how the pool of prompts, $\mathcal{Q}$, is generated.

## 3.3 LEARNING THE VERBALIZER

As discussed, the loss function $\mathcal{L}_t$ as in equation 1 is essentially a weighted sum of the individual loss over the training dataset $\mathcal{D}_{tr}$, *i.e.*

$$\mathcal{L}_t(q_t, M) = \sum_{(x_i, y_i) \in \mathcal{D}_{tr}} w_{ti} \ell(x_i, y_i; q_t, M), \tag{2}$$

where $w_{ti}$ denotes the weight on training data point $i$ for learning weak learner $t$ as determined by ADABOOST; $\ell(x_i, y_i; q_t, M)$ denotes the loss on data point $(x_i, y_i)$ with the parameters set to $q_t$ and $M$. Since we focus on classification tasks, $\ell(\cdot)$ should ideally be the cross-entropy loss. However, the optimization problem in equation 1 is essentially a partition problem, which can easily lead to combinatorial complexity. To derive the tractable solution, we adopt the following strategy. First, solve equation 1 with $\ell(\cdot)$ set to the $\ell_1$ loss, which, though not optimal for the classification task, bears a closed-form solution. Second, further screen the token assignment by maximizing the training set performance. The detailed method is described below.

**Minimizing the $\ell_1$ loss** By replacing the $\ell(\cdot)$ in equation 2 with the $\ell_1$ loss, a closed-form solution can be derived, which can establish a basis for the subsequent steps for deriving a good verbalizer. Formally, let $h_i$ be the one-hot representation of the class label $y_i$, and let $\pi_i = F^*(x_i\|q_t)$ represent the LM output probability with the prompt $q_t$ concatenated. Then, with the $\ell_1$ loss, equation 2 becomes

$$\mathcal{L}_t(q_t, M) = \sum_{(x_i, y_i) \in \mathcal{D}_{tr}} w_{ti} \|h_i - M\pi_i\|_1 = \sum_{(x_i, y_i) \in \mathcal{D}_{tr}} w_{ti} \mathbf{1}^T |h_i - M\pi_i|$$
$$= \sum_{(x_i, y_i) \in \mathcal{D}_{tr}} w_{ti} [(-1)^{h_i}]^T (h_i - M\pi_i). \tag{3}$$

Here, $\mathbf{1}$ represents a all-one column vector of dimension $|\mathcal{Y}|$, and $(-\mathbf{1})^{\boldsymbol{h}_i}$ represents the element-wise power operation. The last equality is because each element of $\boldsymbol{M}\boldsymbol{\pi}_i$ is within $[0, 1]$ and each element of $\boldsymbol{h}_i$ is either 0 or 1, so we can easily remove the absolute sign depending on the actual values of $\boldsymbol{h}_i$.

As shown in equation 3, the loss function is *linear* with respect to $\boldsymbol{M}$, so the optimization in equation 1 becomes a linear optimization problem with linear constraints, which has closed-form corner solutions. For notational brevity, define a score matrix, $\boldsymbol{S}$, as

$$\boldsymbol{S} = \sum_{(\boldsymbol{x}_i, y_i) \in \mathcal{D}_{tr}} w_{ti} \boldsymbol{\pi}_i \big[ (-\mathbf{1})^{\boldsymbol{h}_i} \big]^T, \tag{4}$$

which is the same size as $\boldsymbol{M}$ and is essentially the coefficients multiplied on $\boldsymbol{M}$ in equation 3. Then, we state without detailed derivations that the solution to equation 1 is such that each token is assigned to the class for which it gets the highest score among all the classes, *i.e.*

$$\boldsymbol{M}_{cv} = 1, \quad \text{if } c = \arg\max_{c' \in [|\mathcal{Y}|]} \boldsymbol{S}_{c'v}, \quad \text{and } 0 \text{ otherwise.} \tag{5}$$

Since the $\ell_1$ loss does not generally work well for classification tasks, we empirically find that the verbalizer derived in equation 5 is of limited performance. However, this inspires us that the score matrix, $\boldsymbol{S}$, is a good measure of how well each token should be selected for a class. In the following step, we will further screen the tokens with the help of the score matrix.

**Screening the tokens**  One issue with the verbalizer in equation 5 is that each token has to be assigned to one class, even those tokens that are not well indicative of any classes. Therefore, by removing the non-informative tokens and only retaining the best tokens for each class, we can improve the verbalizer performance. To reduce the computational complexity, we will retain only one token for each class. Specifically, we first identify a candidate set of tokens for each class by choosing the tokens with top-$k$ scores for that class, *i.e.* the top-$k$ elements in $\boldsymbol{S}_{c:}$ for class $c$, where subscript $c$ : denotes the $c$-th row. Then, we evaluate all the possible combinations that include one token from the candidate set for each class (hence $k^{|\mathcal{Y}|}$ combinations in total) and choose the combination that achieves the best training accuracy (weighted by $\{w_{ti}\}$).

## 3.4 Constructing the Prompt Set

To generate the pool of prompts, $\mathcal{Q}$ (step 0 in section 3.2), we adopt the optimization-free method proposed by Gao et al. (2021), which employs the T5 (Raffel et al., 2020) model. Specifically, we first construct a small subset of the training set, denoted as $\mathcal{D}_{gen}$, to induce the prompt generation ($\mathcal{D}_{gen}$ is exactly $\mathcal{D}_{tr}$ in few-shot setting). Then, for each data point $(\boldsymbol{x}_i, y_i) \in \mathcal{D}_{gen}$, we construct an input to the T5 model as `<input><A><label token>` (for sentence-pair classification tasks, the input to T5 becomes `<input1><A><label token><input2>`). Here, `<A>` and `` are mask tokens in T5 representing spans to be filled in. `<input>`, `<input1>` and `<input2>` represent the input text $\boldsymbol{x}_i$. `<label token>` is a pre-defined mapping to convert class labels to tokens in $\mathcal{V}$. For example, positive label ($y_i = 1$) in SST-2 (Socher et al., 2013) dataset is mapped to token `great` while negative label ($y_i = 0$) is mapped to `terrible`. Given this input, the T5 model fills in the spans for `<A>` and ``. The decoding process aims to maximize output probability conditioned on the input over $\mathcal{D}_{gen}$. Then the T5 generated outputs, denoted as `<output A>` and `<output B>` will be converted into prompts and concatenated to the training input text, *i.e.* $\boldsymbol{x}_i \| \boldsymbol{q}$, in the form of `<input><output A>[mask]<output B>` (for sentence-pair tasks, the form becomes `<input1><output A>[mask]<output B><input2>`). As an example, on SST-2 dataset, one of the generated outputs by T5 is `<output A>` = A truly, `<output B>` = movie. Then the input sentence "`I love it.`" will be converted to "`I love it. A truly [MASK] movie`". With a wide beam search width (by default we use 100), we select the top-10 generated prompts according to the log-likelihood to form the prompt pool, $\mathcal{Q}$. All the generated prompts used in our experiments can be found in Table 6 in Appendix C Readers can refer to Gao et al. (2021) for further details. The entire generation process does not involve any optimization over the prompts, and thus is computationally efficient. It is worth noting that the aforementioned approach can be replaced with any other optimization-free prompt generation methods, such as manually creating the prompts, making PROMPTBOOSTING flexible for realistic use.

## 3.5 Ensembling the Weak Learners

We follow the ADABOOST algorithm to ensemble the weak learners. As discussed, each weak learner minimizes a weighted loss over the training set (equation 2). The final prediction is produced by taking a weighted average over the weak classifiers' output. Further details, including how the weights are computed, are shown in Algorithm 1. It is worth mentioning that we can generate many weak learners at a very low computational cost, because we only need to evaluate the LM's output distribution with each of the pre-generated prompts in $\mathcal{Q}$, beyond which no extra forward pass is needed when learning each weak learner. Since the number of pre-generated prompts is small, typically ten in our implementation, the entire learning

---

**Algorithm 1** Model Ensemble in PROMPTBOOSTING

1: **Input:** prompt set $\mathcal{Q} = \bigcup_j \{q_j\}$, LM $F^*(\cdot)$, $\mathcal{D}_{tr}$,
2: **Output:** weak learners $\bigcup_t \{f_t(\cdot)\}$ and their weights $\bigcup_t \{\alpha_t\}$.
3: Set initial data weight to $w_{1i} = 1/|\mathcal{D}_{tr}|, \forall i \in [|\mathcal{D}_{tr}|]$
4: **for** Iteration $t = 1, \ldots, T$ **do**
5:     Randomly draw a prompt $q_t$ from $\mathcal{Q}$
6:     Learn the verbalizer $M_t$ with weight $\{w_{ti}\}$
7:     Set weak learner $t$ to $f_t(\cdot) = M_t F^*(\cdot \| q_t)$
8:     Compute weighted error as
      $err^{(t)} = \sum_{i=1}^{|\mathcal{D}_{tr}|} w_{ti} \mathbf{1}_{y_i \neq f_t(x_i)} / \sum_{i=1}^{|\mathcal{D}_{tr}|} w_{ti}$
9:     Compute the weight on $f_t$ as
      $\alpha_t = \log \frac{1 - err^{(t)}}{err^{(t)}} + \log(|\mathcal{Y}| - 1)$
10:    Adjust dataset weight
      $w_{(t+1)i} = w_{ti} \cdot \exp(\alpha_t \cdot \mathbf{1}_{y_i \neq f_t(x_i)}), \forall i \in [|\mathcal{D}_{tr}|]$
11:    Re-normalize $\{w_{(t+1)i}\}$.
12: **end for**

---

process involves no more than ten forward passes per batch in the training set, no matter how many weak learners are generated.

## 4 EXPERIMENTS

### 4.1 EXPERIMENT SETUP

**Datasets** We conduct experiments on a wide range of tasks including sentiment analysis (SST-2 (Socher et al., 2013) and MR (Pang & Lee, 2005)), topic classification (TREC (Voorhees & Tice, 2000) and AG's News (Zhang et al., 2015)), and natural language inference (SNLI (Bowman et al., 2015), MNLI-m (Williams et al., 2018), RTE (Dagan et al., 2005), and QNLI (Rajpurkar et al., 2016)). The dataset statistics can be found in Table 4 in Appendix A.

**Evaluation Setting** We mainly evaluate the performance of PROMPTBOOSTING in few-shot settings. This is reasonable especially for black-box model tuning scenarios, where the maximum allowed query times may be limited. We randomly sample $k$ examples per class from the original training set to construct a $k$-shot training set $\mathcal{D}_{tr}$ for model training. Following previous work (Gao et al., 2021; Zhang et al., 2021; Sun et al., 2022), we also construct the validation set $\mathcal{D}_{val}$ by randomly sampling another $k$ examples per class from the original training set (i.e., $|\mathcal{D}_{tr}| = |\mathcal{D}_{val}|$). By default we set $k = 16$ for our main experiments. Also, while previous work splits the training and validation sets in this way and we do so for direct comparison, we also explore integrating the validation set into the training set—in a truly few-shot setting, we should make full use of as many examples as we can, and we show this leads to an improvement in performance. As for evaluation, we use the whole testing set. For SNLI (Bowman et al., 2015) and the datasets from the GLUE benchmark (Wang et al., 2018), we use the original validation set for evaluation.

**Backbone Models** In the main experiments, we adopt the widely-used RoBERTa-large model (Liu et al., 2019) for evaluation in order to allow for direct comparison with baselines.

**Baselines** We compare PROMPTBOOSTING with fine-tuning and state-of-the-art black-box tuning methods described below. For reference, we also include white-box prompt-based learning methods that are designed for few-shot setting. Implementation details can be found in Appendix A. **(1) Fine-tuning** is just standard model fine-tuning in a few-shot setting. **(2) LM-BFF** (Gao et al., 2021) is a prompt-based fine-tuning method. In LM-BFF, all input will be transformed using automatically generated prompts. Then the whole model is fine-tuned based on the transformed data. **(3) DART** (Zhang et al., 2021) replaces the discrete prompts in LM-BFF with trainable prompt embeddings, which can reduce the prompt generation cost. **(4) Feature-based MLP** uses LMs as feature extractors, which is a simple yet effective way to train LMs in black-box settings. We take the manual prompt from past work (Gao et al., 2021) and query the LMs to get the the masked token prediction $\mathbf{p}$. Then a 3-layer MLP is trained on the extracted features. **(5) BBT** (Sun et al., 2022) is

Table 1: Performance of proposed PROMPTBOOSTING and baseline methods in few-shot setting ($k = 16$) measured by classification accuracy (%). All methods use RoBERTa-large (Liu et al., 2019) as the backbone LM for fair comparison. Two white-box methods are included for reference including LM-BFF (Gao et al., 2021) and DART (Zhang et al., 2021). Feature-MLP, BBT (Sun et al., 2022), and RLPrompt (Deng et al., 2022) are the main black-box baselines. PROMPTBOOSTING-32 combines both training and validation set for training. Mean accuracy (and standard deviation) is reported over 5 different splits.

| Method | SST-2 | MR | AG's News | TREC | SNLI | MNLI | QNLI | RTE | Avg. |
|---|---|---|---|---|---|---|---|---|---|
| Fine-tuning | 81.4 (3.8) | 82.7 (3.6) | 86.2 (1.4) | 88.8 (2.1) | 48.4 (4.8) | 45.8 (6.4) | 56.3 (1.5) | 54.4 (3.9) | 68.0 |
| LM-BFF | 92.3 (1.5) | 87.4 (0.6) | 87.1 (1.2) | 83.4 (2.7) | 76.5 (2.6) | 68.7 (2.0) | 64.4 (4.6) | 66.6 (6.4) | 78.3 |
| DART | 93.5 (0.5) | 88.2 (1.0) | 86.8 (0.5) | 87.1 (3.8) | 75.8 (1.6) | 67.5 (2.6) | 66.7 (3.7) | 59.0 (2.5) | 78.1 |
| Feature-MLP | 84.9 (3.8) | 82.1 (1.6) | 74.1 (2.0) | 25.3 (2.4) | 57.8 (3.2) | 48.5 (1.8) | 54.4 (4.5) | 55.5 (4.7) | 60.3 |
| BBT | 88.2 (1.7) | 82.8 (2.6) | 81.2 (2.7) | 39.3 (5.2) | 44.7 (4.0) | 42.3 (2.8) | 56.8 (2.0) | 49.1 (3.3) | 60.6 |
| RLPrompt | **90.5 (1.5)** | 86.2 (2.5) | 76.2 (2.7) | 37.3 (3.5) | 42.9 (1.8) | 40.7 (4.7) | 52.1 (2.9) | 52.2 (2.2) | 59.8 |
| PROMPTBOOSTING | 87.6 (3.0) | 84.6 (2.5) | **85.2 (0.9)** | 81.6 (4.0) | 61.3 (3.5) | 52.5 (1.5) | 58.0 (3.3) | 60.0 (5.5) | 71.4 |
| PROMPTBOOSTING-32 | 87.6 (3.3) | 84.7 (2.1) | 84.2 (1.1) | **84.5 (1.4)** | **62.0 (2.7)** | **53.8 (1.2)** | **58.3 (2.8)** | 60.3 (2.4) | 71.9 |

Table 2: Deployment efficiency of proposed PROMPTBOOSTING and baseline methods in few-shot setting ($k = 16$). With all methods use RoBERTa-large (335M parameters) as the backbone LM, some baselines introduce additional parameters, leading to the slight variation in total parameters. Wall time is reported to measure the training time efficiency. Query efficiency is evaluated by #Forward and #Backward, which refer to the number of forward/backward passes per batch during training respectively.

| Method | Trainable param | Total param | AG's News | | | | RTE | | | |
|---|---|---|---|---|---|---|---|---|---|---|
| | | | Acc | Wall Time | #Forward | #Backward | Acc | Wall Time | #Forward | #Backward |
| Fine-tuning | 335M | 335M | 86.2 | 13 min | 100 | 100 | 54.4 | 19 min | 100 | 100 |
| LM-BFF | 335M | 335M | 87.1 | 5min | 32 | 32 | 66.6 | 9 min | 60 | 60 |
| DART | 335M | 335M | 86.8 | 15 min | 30 | 30 | 59.0 | 5 min | 120 | 120 |
| Feature-MLP | 5M | 340M | 74.1 | 0.5 min | 1 | 0 | 55.5 | 0.3 min | 1 | 0 |
| BBT | 25k | 335M | 81.2 | 88 min | 8K | 0 | 49.1 | 52 min | 8K | 0 |
| RLPrompt | 3M | 420M | 77.2 | 117 min | 1K | 0 | 52.2 | 90 min | 1K | 0 |
| PROMPTBOOSTING | <1k | 335M | 85.2 | 8 min | 10 | 0 | 60.0 | 4 min | 10 | 0 |

a black-box tuning method for few-shot learning which employs zeroth-order gradients to optimize the continuous prompts. **(6) RLPrompt** (Deng et al., 2022) models the black-box optimization of discrete prompts as a reinforcement learning problem and adopts Q-learning to find the best prompt.

**Implementation Details** The implementations of standard fine-tuning and feature-based MLP are based on the Huggingface `transformers` library (Wolf et al., 2019). For all other baselines, we use their official implementations and hyper-parameters. Please refer to more details in Appendix A. For our method, we sequentially train weak classifiers and add them to our ensemble—we stop when validation performance plateaus or when we reach the maximum number of weak classifiers.

## 4.2 EVALUATION RESULTS

**Overall Comparison** We first evaluate the effectiveness of PROMPTBOOSTING in a few-shot setting with experiment results in Table 1. Although there is some variance across datasets, PROMPTBOOSTING achieves state-of-the-art performance compared to existing black-box tuning methods.

We emphasize the effectiveness of model ensembling in PROMPTBOOSTING. Firstly, on the SST-2 and MR datasets, which are sentiment analysis tasks, even individual weak learners in PROMPTBOOSTING can achieve 100% accuracy on the training set, making the model ensemble inapplicable (note that AdaBoost cannot ensemble classifiers that achieve 100% accuracy). Therefore, we directly train 10 weak learners using 10 prompts on the unweighted training set and then select the weak learner that performs best on the validation set as the final model. Since the advantage of model ensemble is limited on SST-2 and MR dataset, it is not surprising that PROMPTBOOSTING performs slightly worse than BBT and RLPrompt. However, PROMPTBOOSTING is still better than fine-tuning an-MLP, demonstrating the effectiveness of our proposed verbalizer learning method.

Secondly, on the other 6 datasets, PROMPTBOOSTING consistently outperforms all baselines. PROMPTBOOSTING also outperforms standard fine-tuning on the 4 NLI tasks. It is worth noting that on the TREC dataset, all of the black-box baselines performs very badly except for PROMPTBOOSTING, which even achieves a level of accuracy close to that of white-box methods. One poten-

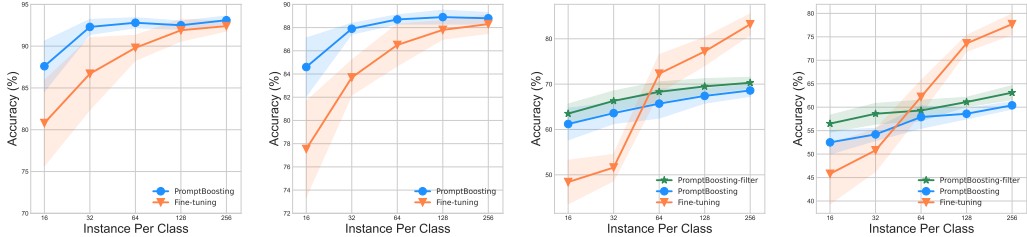

(a) Performance on SST-2    (b) Performance on MR    (c) Performance on SNLI    (d) Performance on MNLI

Figure 2: Model performance as a function of training set size on different datasets. For NLI tasks (SNLI and MNLI), we also include prompt refinement for better performance.

tial reason is that the TREC dataset is harder for prompt-based learning. For example, the manual prompt on the TREC dataset achieves only 32% accuracy Gao et al. (2021). According to our experiments, individual weak learners trained on the unweighted training set using our verbalizer learning method can only achieve 30%-50% accuracy. However, after employing model ensembling, the performance is largely improved, demonstrating the effectiveness of PROMPTBOOSTING.

Finally, we incorporate a variant of PROMPTBOOSTING, namely PROMPTBOOSTING-32, which skips the hyper-parameter tuning and directly integrates the validation set into training. The hyperparameter, i.e., the number of weak classifiers, is determined manually according to its value when the validation set is available. By expanding the training set, one can observe a slight improvement on the performance with lower variance.

**Deployment efficiency** Another concern with black-box model tuning is the deployment efficiency. As we have discussed above, directly adopting zeroth-order gradient optimization techniques suffers from the need to query many times, making it less applicable in realistic scenarios. We visualize the deployment efficiency of different methods in Table 2. AG's News and RTE datasets are adopted due to the average input length (see Table 4). The metrics include parameter efficiency (number of trainable parameters and total parameters), wall time of training, and number of forward/backward passes per batch. In terms of trainable parameters, PROMPTBOOSTING optimizes only less than 1k parameters ($|\mathcal{Y}|*200$) and does not introduce any extra parameters. In contrast, RL-Prompt uses another network, DistilGPT2 (Sanh et al., 2019), in addition to the backbone RoBERTa model and consequently increases the training cost. In terms of wall time, PROMPTBOOSTING improves the efficiency over existing black-box tuning baselines (BBT and RLPrompt) by more than 10 times. The query time is also significantly lower. Only 10 forward passes per batch of training data are required during the training of PROMPTBOOSTING. By contrast, our baselines require thousands of forward passes, which makes them hard to use in realistic scenarios.

**Effect of training data size** We also study the performance of PROMPTBOOSTING as the size of the training set increases (see Figure 2). Note that we still fix $k = 16$ for the validation set regardless of the training set size. Results on AG's News, TREC, QNLI, and RTE dataset are shown in Figure B in Appendix B. The conclusions are in three dimensions. Firstly, on the SST-2 and MR datasets, PROMPTBOOSTING consistently outperforms fine-tuning with lower variance, demonstrating the effectiveness of our method. Secondly, on the AG News and TREC datasets, PROMPTBOOSTING performs worse than fine-tuning. A similar phenomenon also exists in past work (Gao et al., 2021), where even a white-box prompt-based few-shot learning method can achieve performance that is at most only comparable with fine-tuning. However, we remark that our method still maintains large advantages compared to all black-box baseline methods and achieves highly usable performance. Finally, as the amount of training data increases, the performance of fine-tuning improves and gradually outperforms our method on the four NLI datasets. This finding is possibly due to the fact that pre-trained LMs before fine-tuning are not good at tasks involving sentence pairs.

**Refinement of prompts** The performance of the weak learner in PROMPTBOOSTING directly depends on the prompt. As has been shown in previous work, different prompts have significant influence on the performance of prompt-based methods (Shin et al., 2020; Gao et al., 2021). However,

Table 3: Performance of PROMPTBOOSTING with different numbers of prompts in few-shot setting ($k = 16$).PROMPTBOOSTING-$d$ means top-$d$ prompts (sorted according to the beam search score) are used for model training. Mean accuracy (and standard deviation) is reported over 5 different splits.

|                   | SST-2      | MR         | AG's News  | TREC       | SNLI       | MNLI       | QNLI       | RTE        | Avg. |
|-------------------|------------|------------|------------|------------|------------|------------|------------|------------|------|
| PROMPTBOOSTING-1  | 86.1 (1.0) | 85.1 (5.0) | 73.3 (3.7) | 41.3 (4.3) | 53.4 (4.0) | 49.5 (3.5) | 58.0 (2.4) | 56.5 (5.7) | 62.9 |
| PROMPTBOOSTING-5  | 88.8 (1.9) | 87.9 (1.6) | 83.5 (4.2) | 78.0 (2.5) | 59.1 (3.5) | 50.9 (4.8) | 56.5 (2.1) | 57.0 (4.5) | 70.2 |
| PROMPTBOOSTING-10 | 87.6 (3.0) | 84.6 (2.5) | 85.2 (0.9) | 81.6 (4.0) | 61.3 (3.5) | 52.5 (1.5) | 58.0 (3.3) | 60.0 (5.5) | 71.4 |
| PROMPTBOOSTING-20 | 88.1 (2.6) | 84.0 (2.3) | 86.4 (1.3) | 81.9 (2.7) | 60.8 (3.9) | 55.2 (1.2) | 57.0 (4.4) | 57.1 (3.2) | 71.3 |

in PROMPTBOOSTING, the prompts are fixed and will not be optimized during training. Therefore, we consider a simple yet effective way to improve the performance through prompt refinement. Specifically, because we automatically generate 100 prompts for each dataset but only use 10 of them, we may select top-10 prompts following some heuristics to improve the quality of prompts. Before training, we first evaluate the performance on the validation set by training a weak classifier using the method in Section 3.3 on the unweighted few-shot training set. Then we construct the prompt pool by selecting the top-10 prompts according to the accuracy of the corresponding weak learner on validation set. Please note that the few-shot setting makes the refinement process very efficient. Later on, PROMPTBOOSTING is trained using the refined prompts. We mainly evaluate the effectiveness of the prompt refinement on SNLI, MNLI, and QNLI dataset where the gap between PROMPTBOOSTING and standard fine-tuning is relatively large with the increase of training data. Experiment results can be found in Figure 2. One can observe consistent improvements on few-shot performance across three NLI tasks, especially on QNLI dataset where the performance of PROMPTBOOSTING was far from satisfactory without prompt refinement. Overall, the prompt refinement leads to a trade-off between training cost and model performance.

**Effect of the number of prompts** In our main experiments, we use 10 prompts by default. Intuitively, a large prompt pool increase the diversity of weak classifiers which could improve the performance. However, the training/inference cost will also increase if more prompts are included for model training. We empirically study the relationship between the number of prompts and the model performance in Table 3. In general, more prompts benefit the performance for most datasets (except QNLI). We highlight the effectiveness of multiple prompts on AG's News and TREC dataset, on which the performance becomes better and more stable. As we have discussed in the few-shot experiments in Table 1, individual prompt performs very bad on TREC dataset. This is also proved by PROMPTBOOSTING-1 that only achieves 41.3% accuracy. However, by using our prompt ensemble framework, the performance can be boosted to 84.6% when 10 prompts are provided. Finally, the performance improvement is relatively small when the number of prompts increase from 10 to 20, implying that 10 prompts should be good enough for PROMPTBOOSTING.

**Full data training** Due to its efficiency, PROMPTBOOSTING can also generalize to full data training instead of just the few-shot setting. We compare with fine-tuning on the entire training dataset for SST, MR, TREC, and RTE. Experimental results can be found in Table 5 in the Appendix B.

## 5 CONCLUSION

In this paper, we propose PROMPTBOOSTING, an effective black-box model tuning framework. Without access to the parameters and gradients of pre-trained LMs, PROMPTBOOSTING can adapt LMs for various downstream tasks. The efficient weak learner construction method, together with the ADABOOST model ensemble algorithm, makes PROMPTBOOSTING achieve state-of-the-art performance in black-box tuning setting with at least 10x run-time efficiency.

For future directions, we will explore how to generalize PROMPTBOOSTING beyond the classification tasks to generation tasks. Also, we will study how to combine the prompt ensemble idea in PROMPTBOOSTING with gradient-based optimization and further improve the performance of existing prompt-based learning methods.

## REPRODUCIBILITY STATEMENT

The authors have made great efforts to ensure the reproducibility of the experiment results in the paper. Firstly, the experiment settings, evaluation metrics, benchmarks, *etc.* are explained in detail in Section 4.1. In summary, eight public datasets with clear references are used for evaluation in few-shot setting. Secondly, the implementation details and hyper-parameters of our method and baselines are clearly presented in Section B and Appendix A. We also include all prompts that are used for our method in Appendix C. Thirdly, all experiments are based on 5 runs with different random seeds to improve the reliability. Finally, the pre-released code for the proposed method is also included in the supplemental material.

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

Table 4: The dataset statistics. $|\mathcal{Y}|$ is the number of classes, Avg.#W is the average number of words in the input, and #Train/#Test refer the number of examples in training/testing dataset.

| Category | Dataset | $|\mathcal{Y}|$ | Avg. #W | #Train | #Test |
|---|---|---|---|---|---|
| single sentence | SST-2 | 2 | 17 | 6920 | 872 |
| | MR | 2 | 20 | 8662 | 2000 |
| | AG's News | 4 | 47 | 120000 | 7600 |
| | TREC | 6 | 10 | 5452 | 500 |
| sentence pair | SNLI | 3 | 22 | 549367 | 9842 |
| | MNLI | 3 | 33 | 392702 | 9815 |
| | QNLI | 3 | 41 | 104743 | 5463 |
| | RTE | 3 | 59 | 2490 | 277 |

Table 5: Performance of full data training

| | SST-2 | MR | TREC | RTE |
|---|---|---|---|---|
| Fine-tuning | 95.5 (0.4) | 91.5 (0.6) | 97.2 (0.2) | 81.9 (1.1) |
| PROMPTBOOSTING | 94.1 (0.3) | 89.7 (0.4) | 90.5 (1.2) | 71.7 (2.0) |

## A  IMPLEMENTATION DETAILS

**Dataset Statistics**   The dataset statistics can be found in Table 4. For fair comparison, the few-shot training/validation/testing split generation is strictly following the implementation of Gao et al. (2021).

**Training of baselines**   For standard fine-tuning, we adopt the Huggingface `transformers` library (Wolf et al., 2019) to load RoBERTa-large backbone model and use its `Trainer` for fine-tuning. The learning rate is set to 1e-5. We use AdamW optimizer as the optimizer and the learning rate linearly decays to 0. The training batch size is set to 16 and total training epochs is 100. For Feature-MLP method, we use a three-layer MLP with hidden dimension 100. The learning rate is set to 1e-3 without learning rate decay. We also train the MLP for 100 epochs. For other baselines, we use their the official implementation with default hyper-parameters including LM-BFF (Gao et al., 2021), DART (Zhang et al., 2021), BBT (Sun et al., 2022), and RLPrompt (Deng et al., 2022). For RLPrompt, because of its low efficiency, we set its training epochs to 1000 instead of the 12000 used in their paper. This is reasonable since it takes nearly 2 hours for RLPrompt to finish 1000 epochs of optimization.

## B  ADDITIONAL EXPERIMENTS

**Performance on full dataset**   The high efficiency of PROMPTBOOSTING makes it possible to generalize to medium-sized datasets. We evaluate the performance of PROMPTBOOSTING on SST-2, MR, TREC, and RTE datasets. We sample 10% of the original training set to construct the validation set and use the original validation set for testing if the labeled test set is unavailable. The experiment results can be found in Table 5. PROMPTBOOSTING achieves comparable performance with standard fine-tuning on SST-2 and MR datasets, which is impressive given the fact that PROMPTBOOSTING has no access to the parameters and gradients of the LM. For TREC dataset, standard fine-tuning outperforms PROMPTBOOSTING, but we still remark that the performance is still highly usable in black-box setting. Finally, the gap between PROMPTBOOSTING and fine-tuning is relatively large on RTE dataset, which is consistent with our previous discovery that it seems pre-trained LMs are not good at sentence pair classification tasks before fine-tuning.

**Effect of training data size**   For AGNews, TREC, QNLI, and RTE datasets, we shown the performance of PROMPTBOOSTING as the size of the training set increases in Figure B.

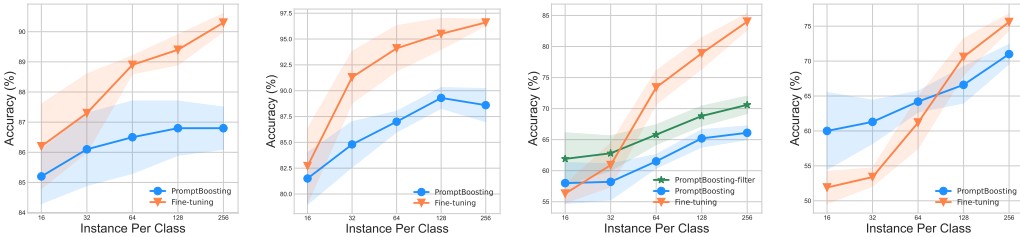

(a) Performance on AG  (b) Performance on TREC (c) Performance on QNLI  (d) Performance on RTE

Figure 3: Model performance as a function of training set size on different datasets. For QNLI dataset, we also include prompt refinement for better performance.

Table 6: Prompts used by PROMPTBOOSTING on different datasets.

| | SST-2 | MR |
|---|---|---|
| 1 | [Input] It's [MASK]. | [Input] It's [MASK]. |
| 2 | [Input] A [MASK] movie. | [Input] It's [MASK]! |
| 3 | [Input] A [MASK] film. | [Input] A [MASK] piece of work. |
| 4 | [Input] A [MASK] piece of work. | [Input] It's [MASK]. |
| 5 | [Input] A truly [MASK] film. | [Input] A [MASK] waste of time. |
| 6 | [Input] This is [MASK]. | [Input] A truly [MASK] film. |
| 7 | [Input] It was [MASK]. | [Input] I thought it was [MASK]. |
| 8 | [Input] A [MASK] waste of time. | [Input] It's just [MASK]. |
| 9 | [Input] It's [MASK]! | [Input] A truly [MASK] movie. |
| 10 | [Input] A truly [MASK] movie. | [Input] The film is [MASK]. |

| | AG's News | TREC |
|---|---|---|
| 1 | [Input] This entry was posted in [MASK]. | [Input] What is [MASK]? |
| 2 | [Input] U.S. [MASK] News. | [Input] What is the [MASK]? |
| 3 | [Input] U.S. [MASK]. | [Input] What [MASK]? |
| 4 | [Input] This entry was posted in [MASK] News. | [Input] The [MASK]. |
| 5 | [Input] The [MASK] Journal reports. | [Input] See [MASK]. |
| 6 | [Input] The [MASK] Journal has more. | [Input] Which [MASK]? |
| 7 | [Input] Read more at [MASK] News Now. | [Input] The [MASK]? |
| 8 | [Input] The New York Times [MASK]. | [Input] Full [MASK]. |
| 9 | [Input] The New York Times [MASK] Report. | [Input] How many [MASK]? |
| 10 | [Input] Read more at[MASK] Insider. | [Input] 1.[MASK]. |

| | SNLI | MNLI |
|---|---|---|
| 1 | [Input1]. [MASK], [Input2] | [Input1]. [MASK], [Input2] |
| 2 | [Input1]. [MASK]. [Input2] | [Input1]. [MASK], but [Input2] |
| 3 | [Input1]. [MASK] and [Input2] | [Input1]. [MASK]. [Input2] |
| 4 | [Input1]. [MASK], but [Input2] | [Input1]! [MASK], [Input2] |
| 5 | [Input1]. [MASK]: [Input2] | [Input1]. [MASK]. But [Input2] |
| 6 | [Input1]. [MASK] one of [Input2] | [Input1]? [MASK], [Input2] |
| 7 | [Input1]. [MASK]... [Input2] | [Input1]. [MASK] and [Input2] |
| 8 | [Input1]. [MASK], just [Input2] | [Input1]. [MASK], and [Input2] |
| 9 | [Input1]. [MASK] it is [Input2] | [Input1]. [MASK] but [Input2] |
| 10 | [Input1]. [MASK]; [Input2] | [Input1]. [MASK]... [Input2] |

| | QNLI | RTE |
|---|---|---|
| 1 | [Input1]? [MASK], [Input2] | [Input1]. [MASK], [Input2] |
| 2 | [Input1]? [MASK], but [Input2] | [Input1]. [MASK]. [Input2] |
| 3 | [Input1]? [MASK]. [Input2] | [Input1]. [MASK], but [Input2] |
| 4 | [Input1]? [MASK]. But [Input2] | [Input1]. [MASK] and [Input2] |
| 5 | [Input1]? [MASK]. In fact, [Input2] | [Input1]. [MASK]: [Input2] |
| 6 | [Input1]? [MASK]; [Input2] | [Input1]. [MASK], the [Input2] |
| 7 | [Input1]? [MASK]. However, [Input2] | [Input1]. [MASK]; [Input2] |
| 8 | [Input1]? [MASK], and [Input2] | [Input1]. [MASK]-[Input2] |
| 9 | [Input1]? [MASK]: [Input2] | [Input1]. [MASK], and [Input2] |
| 10 | [Input1]. [MASK], [Input2] | [Input1]. [MASK] but [Input2] |

## C  GENERATED PROMPTS

In this section, we visualize the prompts we used in our experiments for each dataset in Table 6. Regardless of different few-shot training/validation splits, we use the same 10 prompts for model training.

