# OpenReview forum: "PromptBoosting: Black-Box Text Classification with Ten Forward Passes"
_ICLR.cc/2023/Conference — Submitted to ICLR 2023_

### Official Review · Reviewer_cHte · 2022-10-23

**Confidence:** 2
**Clarity, Quality, Novelty And Reproducibility:** 1. Very clearly written paper. Well m…
**Correctness:** 3
**Technical Novelty And Significance:** 2
**Empirical Novelty And Significance:** 2
**Recommendation:** 6

**Strength And Weaknesses:**

- Strength
 1. Clearly written, well motivated
 2. Competitive results
 3. Reasonable methods

-Weakness
1. I do believe the method will work but somehow I don't see a further discussion on the best scheme when boosting is used.
2. not so many black-box method is deployed in this direction, and there is no thorough discussion of the potential applicability of previous black-box methods. That is, I believe zero-order method has a broader literature but this paper didn't connect with that well.

**Summary Of The Paper:**

This paper deals with the problem of prompt-style finetuning when gradient information is not available. The solution is using adaboost to quickly learn prompt-style learners. Experimental results on standard benchmark showed competitive results over previous black-box methods.

**Summary Of The Review:**

Overall I like the research problem and the method proposed. Not very difficult or sophisticated solution but sounds working. I hope authors can add further studies to validate the root cause. Disclaimer is that I am not particularly familiar with fine-tuning under black-box methods. I do study black-box methods under other setups but not sure if this paper covers all recent works, and if recent works covered all possible black-box methods.

---

### Official Review · Reviewer_QZLQ · 2022-10-25

**Confidence:** 3
**Correctness:** 3
**Technical Novelty And Significance:** 3
**Empirical Novelty And Significance:** 3
**Recommendation:** 6

**Clarity, Quality, Novelty And Reproducibility:**

See weaknesses. Besides, the code is without running scripts. How to run the code?

**Strength And Weaknesses:**

Strength：

1.This paper proposes a novel approach that does not rely on searching for an optimal prompt and which can thus drastically improve the computational efficiency over the existing method. In my opinion, the proposed approach is well-written and well-motivated.

2.Comprehensive experimental results show that the proposed approach is efficient and effective.

Weaknesses:

From my point of view, I think the prompt set may have a big influence on the performance. As the paper mentioned, "the aforementioned approach can be replaced with any other optimization-free prompt generation methods," so can you provide the performance with other prompt generation methods?


**Summary Of The Paper:**

This paper proposes PROMPTBOOSTING, a novel black-box prompt learning approach that does not rely on searching for an optimal prompt and which can thus drastically improve the computational efficiency over the existing method. Specifically, the proposed approach obtains a small pool of prompts via a gradient-free approach and then constructs a large pool of weak learners by pairing these prompts with different elements of the LM’s output distribution. These weak learners are then ensembled using the AdaBoost algorithm. The learning process requires only a small number of forward passes per batch and no backward pass. Experiments show that PROMPTBOOSTING achieves state-of-the-art performance in multiple black-box few-shot classification tasks and matches or outperforms full fine-tuning in both few-shot and standard learning paradigms while training 10x faster than existing black-box methods. Overall, this paper is well-written and well-motivated.

**Summary Of The Review:**

Overall, I think this paper is well-written and well-motivated. Some small issues are the lack of detailed analysis of different prompt templates. I think black-box tuning is a promising solution for foundation models, and this work can help inspire future work.

---

### Official Review · Reviewer_jDoU · 2022-10-29

**Confidence:** 4
**Clarity, Quality, Novelty And Reproducibility:** 1
**Correctness:** 3
**Technical Novelty And Significance:** 3
**Empirical Novelty And Significance:** 3
**Recommendation:** 6

**Strength And Weaknesses:**

Strength:

1: Black box setting for LLM with prompt is an interesting direction and would have lots of applications, especially for the cases the LLM is too large.

2: Paper is well written and the proposed idea makes sense.

3: Comparing with many methods and datasets, and showing better results than other methods.

Weakness:

1: The scope of the work seems limited. This work seems can only be used for text classification (due to the Verbalizer step which needs the end task to be classification task only). Therefore a large amount of use cases for LLM with few shot learning settings such as summarization, etc cannot be directly applied with the proposed method.

2: Missing some ablation studies. Such as: the prompts sets are generated by optimization-free method proposed by Gao et al. (2021). How about using other prompt sets, maybe just original training examples as a simple baseline? And the adaboost is used for ensemble weak learners, how about use other ensemble methods?

3: Some of the experimental results need further explanation.
 3.1 Is the training time (wall time) including prompt set generating time? Or just the time for training weak learners? Could you provide time for each step of the proposed method?

3.2 In Table 2, DART and LM-BFF need lots of backward and forward passes of the model, why the training time is close to the proposed method? Does it mean the forward and backward time is not the bottleneck?

3.3. And also finetuning results are almost always worse than the proposed method (and DART and LM-BFF), which is beyond my expectation. It is better to have some explanation for that.

**Summary Of The Paper:**

This paper proposed PromptBoosting, which works on black box setting for text classification with LLM and prompt. The idea is to build a set of weak learners and each of them is associated with a prompt (raw text) and the final model is trained with adaboost over these weak learners. The resulting weak learners achieve state-of-the-art performance in multiple black-box few-shot text classification tasks, and matches or even outperforms full fine-tuning in both few-shot and standard learning.

**Summary Of The Review:**

The paper is well written and easy to read. The idea is simple and makes sense. There are lots of experiment provided in the paper showing good performance of the proposed method for text classification.  However I am mostly concerned about the limit of the scope of the work (limited to text classification only).

---

### Official Review · Reviewer_QXXp · 2022-10-31

**Confidence:** 4
**Correctness:** 3
**Technical Novelty And Significance:** 2
**Empirical Novelty And Significance:** 2
**Recommendation:** 6

**Clarity, Quality, Novelty And Reproducibility:**

Clarity of the paper is good that ideas are clearly conveyed.

The novelty is slim, as it sounds like a direct application of adaboost on the prompting methods.

I did not check the code so I cannot judge the reproducibility.

**Strength And Weaknesses:**

Strength:

1. The method is very intuitive in applying the boosting idea to the prompt-based methods such as Gao et al. (2021). Also, the paper is well written. So in general, this paper is easy to understand.
2. Learning the verbalizer has a closed form solution, which makes it very efficient.

Weaknesses:

1. The novelty of the paper seems slim. It seems like a direct application of adaboost on the existing prompting method Gao et al. (2021).
2. The evaluation is not convincing.
    2.1. First of all, according to the implementation, this method can only be applied to text classification tasks, making it a bit narrow.
    2.2. Besides, while there a quite a few text classification tasks, the authors only select 8 without giving the rational why they are selected. In contrast, Gao et at. (2021) evaluates their method on 16 tasks, mostly from (Super)GLUE. I suggest  the author at least extend the evaluation on more representative tasks to show the effectiveness is indeed universal.
    2.3. Finally, there should be some simple baselines to compare with, e.g., prompt ensemble with majority vote. But they are never shown up in the experiments.


**Summary Of The Paper:**


This paper proposes a prompting-based method for building a text classifier from language models (LMs), which is called PromptBoosting. The idea of the proposed method is to first obtain a small pool of prompts via gradient-free approach, and then contract a large pool of weak learners by pairing the prompts with different elements of the LM’s output distribution. Then the adaboost algorithm is applied to ensemble the weak learners. Empirical studies claim that PromptBoosting can achieve state-of-the-art performance in different few-shot classification tasks while being 10x faster than existing black-box methods.

**Summary Of The Review:**

The paper applies the adaboost algorithm to the existing prompting methods to get the better text classifier with pretrained LM. While the experimental results look promising, the lack of novelty and comprehensiveness of experiments are two concerns.

---

### Decision · Program_Chairs · 2023-01-20

**Decision:**

Reject

**Justification For Why Not Higher Score:**

- Novelty slim over LM-BFF + Adaboost
- Limited applicability to only text classification due to use of verbalizer.
- Despite motivating the method as "cost of training and inference in large-scale LMs grows", all experiments were conducted on 300M parameter model. Transfer of method to larger sized model is not clear.

**Justification For Why Not Lower Score:**

N/A

**Metareview: Summary, Strengths And Weaknesses:**

The paper attempts to improve few-shot learning and make it more efficient. In this regards, authors adopt an approach based on prompting large language models. The idea is to first generate a small pool of prompts via gradient-free approach, and then construct a large pool of weak learners by pairing the prompts with different elements of the LM’s output distribution. Empirical studies provided seem to indicate effectiveness of the approach. All the reviewers found the paper easy to follow and the approach to be intuitive, but unfortunately at the same time consider the novelty to be slim over a combination of LM-BFF + Adaboost. Also reviewers pointed out limited applicability of the proposed method to only text classification due to use of verbalizer. We thank the authors to engage with reviewers during the discussion phase towards improving the paper. Many new results were added during the discussion phase, including omitted datasets providing a more comprehensive dataset coverage with prior work and experiments on larger model sizes. These studies are crucial to determine the effectiveness claim, but still questions around it remain. For example, in comparison to OPT, the baseline number, e.g. for RTE, seems weaker than original paper in zero-shot setting. Incorporating all the new results and reviewer feedback would require a fair amount of rewriting and thus warrant another round of reviewing.

**Summary Of Ac-Reviewer Meeting:**

- Reviewers maintained their positions and didn't feel strongly in either way
- Reviewers mostly converged on the score to on the border
- The author response only partially addressed reviewer concerns like novelty as compared to adaboost or limited applicability